# Continuous-wave electrically pumped multi-quantum-well laser based on group-IV semiconductors

Lukas Seidel [1,12] ✉, Teren Liu[2,12], Omar Concepción [2], Bahareh Marzban[3,9], Vivien Kiyek [2,10], Davide Spirito [4,11], Daniel Schwarz [1], Aimen Benkhelifa [1], Jörg Schulze[5], Zoran Ikonic [6], Jean-Michel Hartmann[7], Alexei Chelnokov [7], Jeremy Witzens [3], Giovanni Capellini [4,8], Michael Oehme[1], Detlev Grützmacher [2] & Dan Buca [2] ✉

Over the last 30 years, group-IV semiconductors have been intensely investigated in the quest for a fundamental direct bandgap semiconductor that could yield the last missing piece of the Si Photonics toolbox: a continuous-wave Si-based laser. Along this path, it has been demonstrated that the electronic band structure of the GeSn/SiGeSn heterostructures can be tuned into a direct bandgap quantum structure providing optical gain for lasing. In this paper, we present a versatile electrically pumped, continuous-wave laser emitting at a near-infrared wavelength of 2.32 μm with a low threshold current of 4 mA. It is based on a 6-periods SiGeSn/GeSn multiple quantum-well heterostructure. Operation of the micro-disk laser at liquid nitrogen temperature is possible by changing to pulsed operation and reducing the heat load. The demonstration of a continuous-wave, electrically pumped, all-group-IV laser is a major breakthrough towards a complete group-IV photonics technology platform.

Si-based photonic integrated circuits (PICs) in which optically active components are monolithically integrated on a silicon chip are transforming the next generation of information and communication technology infrastructure[1]. Due to its excellent optical performance, inherent flexibility, and low cost, the Si photonic (SiPh) technology has yielded active and passive components that are already widely employed in a wealth of applications, ranging from datacom to sensing in biological and environmental detection systems. Moreover, SiPh develops as a promising technology platform for cryogenic applications in the emerging fields of integrated quantum technologies, optical computing, and artificial intelligence related technologies[1-5]. Along this route, advances have been made in key SiPh components, including high-performance modulators, photodetectors, and waveguides[4,6-10]. Nonetheless, an efficient, electrically pumped light source that can be manufactured using solely group-IV semiconductors still remains elusive. Although other approaches like bonding or microprinting III-V lasers on Si reached manufacturing level[11], the ability to monolithically fabricate laser devices with Si

[1]Institute of Semiconductor Engineering, University of Stuttgart, 70569 Stuttgart, Germany. [2]Peter Gruenberg Institute 9 (PGI-9) and JARA-Fundamentals of Future Information Technologies, Forschungszentrum Juelich, 52428 Juelich, Germany. [3]Institute of Integrated Photonics, RWTH Aachen, 52074 Aachen, Germany. [4]IHP - Leibniz Institut für innovative Mikroelektronik, 15236 Frankfurt (Oder), Germany. [5]Chair of Electron Devices, Friedrich-Alexander-Universität Erlangen-Nürnberg, 91058 Erlangen, Germany. [6]Pollard Institute, School of Electronic and Electrical Engineering, University of Leeds, Leeds LS2 9JT, UK. [7]Université Grenoble Alpes, CEA, LETI, 38054 Grenoble, France. [8]Department of Sciences, Università Roma Tre, 00146 Roma, Italy. [9]Present address: Institute for Quantum Electronics, ETH Zürich, 8093 Zürich, Switzerland. [10]Present address: Institute of Energy Materials and Devices (IMD-2), Forschungszentrum Juelich, 52428 Juelich, Germany. [11]Present address: BCMaterials, Basque Center for Materials, Applications and Nanostructures, UPV/EHU Science Park, 48940 Leioa, Spain. [12]These authors contributed equally: Lukas Seidel, Teren Liu. ✉e-mail: lukas.seidel@iht.uni-stuttgart.de; d.m.buca@fz-juelich.de

technology processing steps would massively simplify the integration and lead to large-scale miniaturization. Besides manufacturing-related aspects, the GeSn lasers presented here give access to new functionalities and applications in the 2–4-μm wavelength range where the availability of lasers based on other materials is very reduced[12,13]. GaSb-based III-V material systems like GaInAsSb/AlGaAsSb have been successfully grown on Si and open up a similar wavelength range as GeSn[14] in the mid-IR, that is particularly important for molecular sensing. However, it does not offer the advantage of all-group-IV semiconductors and presents much more significant epitaxy and contamination issues[15]. Moreover, as a compound semiconductor, it is also subjected to the formation of antiphase domain boundaries, the suppression of which requires growth on patterned Si. An ideal on-chip light source based on Si needs to meet the following criteria: (i) low electrical current injection using bias and currents compatible with chip-scale drivers; (ii) continuous-wave (CW) or pulsed operation, depending on the application; (iii) temperature-stable operation in the targeted application range; and (iv) compatibility with standard complementary metal–oxide–semiconductor (CMOS) Si processing technology.

The successful fabrication of such a device would allow to leverage the mature CMOS technology and enable the integration of even more complex functionality into the SiPh platform, beneficial for e.g., nanoelectronics[16,17], emerging neuromorphic[7] or quantum-computing[8] hardware platforms operating at cryogenic temperatures[18]. For instance, a fully integrated SiPh cryogenic circuitry may overcome the data bottleneck in spin-qubit and superconductor-based quantum-computing systems, requiring unconventional short-reach optical interconnects[19].

The newly developed GeSn and SiGeSn semiconductor material system offers an essential advantage over other group-IV semiconductors: its lattice can be designed, by properly selecting the alloy composition and epitaxial strain, so that the material is turned into a fundamental direct bandgap semiconductor[20–22]. This unique property makes the (Si)GeSn system very attractive for photonic applications[10,23,24]. Major milestones towards the realization of a group-IV laser were reached in recent years. The first breakthrough was the demonstration of pulsed, optically pumped lasing, with initial operating temperatures limited to 90 K from an edge-emitting cavity. Improvements in the active material quality and new device concepts led to room temperature operation under optical pumping and reduced thresholds[25–27]. However, despite all these efforts, to date, only one publication reported CW operation, under optical pumping in tensile strained GeSn disk lasers[21]. Another significant finding was a tenfold laser threshold reduction achieved using GeSn/SiGeSn multiquantum wells (MQWs)[28,29] as active layers. Electrically pumped lasing was reported based on SiGeSn/GeSn double heterostructures with an almost micron-thick gain material layer and solely under nanoseconds pulsed operation with high current injection[30,31]. Thus, two major key characteristics have still to be demonstrated: electrically pumped operation in CW regime and at low current injection.

In this work, both milestones are reached through SiGeSn/GeSn MQW disk-cavity laser diodes designed for cryogenic operation (4–77 K). The electrically pumped microdisk laser operates from CW to short pulse conditions, at injection currents as small as 4 mA.

## Results

### The laser heterostructure and the design principle

The GeSn/SiGeSn epitaxial heterostructure is grown in an industrial reduced-pressure chemical-vapor-deposition reactor (RP-CVD) on a 200-mm Si(100) wafer buffered with a 2.5-μm-thick Ge virtual substrate (VS) in-situ boron-doped to $10^{18}$ cm$^{-3}$ [32–34]. A partially relaxed 200 nm Ge$_{0.9}$Sn$_{0.1}$ buffer is first grown, that sets a larger in-plane lattice constant for the following heteroepitaxy of the optically active MQW stack. The MQW consists of six periods of 20 nm thick Si$_{0.06}$Ge$_{0.83}$Sn$_{0.11}$

barriers and 40 nm thick Ge$_{0.885}$Sn$_{0.115}$ wells. On top of the MQW stack, an n-type Si$_{0.06}$Ge$_{0.83}$Sn$_{0.11}$ electron injection layer is grown. The high quality of the epitaxy and the atom distribution is shown in Fig. 1a, where a scanning transmission electron micrograph (STEM) of the heterostructure is displayed together with its Si, Sn, and Ge elemental distributions obtained from energy-dispersive X-ray mapping.

The Sn concentration of 11.5 at.% was low enough to offer a high crystalline quality while being high enough to decrease the Γ-valley conduction band (CB) energy $E_\Gamma$ and provide a sufficient energy separation $\Delta E_{\Gamma\text{-L}}$ (here called directness) from the indirect L-valley $E_L$, which is required for high optical gain. The Si content of 6 at.% is one of the lowest possible values yielding a Type I band alignment (i.e., confinement for both electrons and holes), given that a pseudomorphic Si$_y$Ge$_{1-x-y}$Sn$_x$ layer grown on a relaxed GeSn buffer becomes tensile strained[35].

The heterostructure differs from a typical MQW laser stack in the design of both the optically active region and the injection layers with their doping mechanism. For the MQWs design: (i) the well thickness was chosen to give a minimum level of quantization, just enough to change from a 3D to a 2D density of states (DOS) while preserving the directness $\Delta E_{\Gamma\text{-L}}$ inside the GeSn wells. Indeed, owing to the large difference in their effective masses[32], the energy of quantized Γ-valley states increases much more than that of L-valley states with decreasing well thicknesses, compromising the directness of the material. (ii) The purpose of the well/barrier heterostructure is to induce spatial confinement of carriers. The barriers are thin enough to provide an overlap of wave functions from adjacent wells, allowing carrier transport by tunneling. At the same time, they are not so thin as to destroy spatial confinement and quantization. (iii) The design of the hole injection layer was modified: instead of being doped during the growth, the layer, non-intentionally doped, benefits from the dense misfit dislocation network at the GeSn heterointerface with the Ge-VS. The acceptor-like states of the network have an estimated effective doping density of $5 \times 10^{17}$–$1 \times 10^{18}$ cm$^{-3}$ [36].

### The laser cavity

The laser cavity has been designed as an under-etched microdisk (Fig. 1b). Fabrication details can be found in the Methods section. Scanning electron microscopy (SEM) images of the device at different fabrication stages are shown in Fig. 1c (insets $c_1$–$c_3$). The under-etch of the microdisk results in a high refractive index contrast and strong confinement of light at the rim of the GeSn disk, where whispering gallery modes (WGMs) of the cavity form. It also enables to elastically relax the residual epitaxial strain present in the as-grown heterostructure. For all fabricated microdisk cavities, the extent of the under-etch was chosen to be 900 nm, in order to maintain a large Ge pillar for heat dissipation. The band structure in the strain relaxed under-etched WGM region is shown in Fig. 1d. While material parameters for band structure calculations are known and experimentally validated for GeSn binaries, the bowing parameter $b_{SiSn}$ in SiGeSn ternaries depends on the Ge concentration[32,37,38]. To determine it for our barrier material, the photoluminescence emission of the top Si$_{0.06}$Ge$_{0.83}$Sn$_{0.11}$ alloy was measured. A fit of the experimental data yielded $b_{SiSn} = 19$ eV, a value which was used for electronic band structure calculations using the 8-band $k \cdot p$ method[35,39]. A type I band alignment is obtained (Fig. 1d) with offsets at the well-barrier interface of ~10 meV for electrons and ~40 meV for holes, providing carrier confinement up to ~110 K. Quantization induces an energy level splitting between the first two Γ-valley sub-bands of ~1-2 meV, so that multiple sub-bands will be populated beyond ~20 K. Similarly, multiple hole sub-bands are populated.

### The CW laser

The devices were placed in a cryostat and their light emission analyzed using a Fourier-transform spectrometer in continuous-scan mode with a maximum resolution of 0.2 cm$^{-1}$ for CW pumping

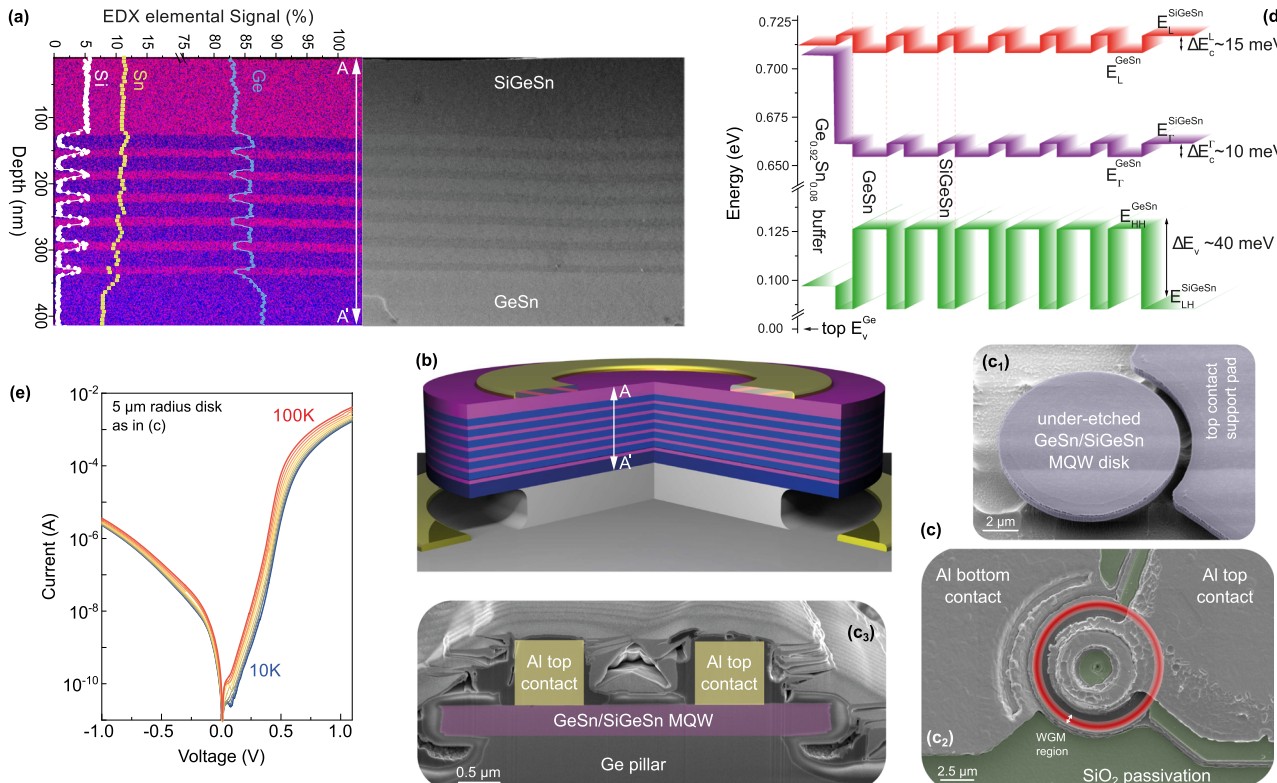

**Fig. 1 | SiGeSn/GeSn multi-quantum-well structure. a** Cross-sectional TEM image of the MQW together with Ge and Sn EDX elemental 2D map overlaid with the line cut Si, Ge, and Sn elemental concentrations. **b** Schematic 3D image of an under-etched microdisk laser without passivation. **c** SEM micrographs showing a (c1) microdisk after under-etching, (c2) top-view with highlighted WGM region, and (c3) cross-section of a fabricated device. **d** Electronic band diagram of the MQW in the optically active under-etched region. **e** Current-voltage characteristics of the microdisk laser for temperatures ranging from 10 K to 100 K.

experiments (see Methods). The I-V characteristics of a laser diode are presented in Fig. 1e, showing rectifying behavior in the measured temperature range from 10 to 100 K. A series of spectra recorded from a microdisk with a 5-μm-radius under different injection currents is presented in Fig. 2a. The optical emission evolves from spontaneous to stimulated emission and back again to spontaneous emission as the injection current is increased. This is shown by a single, sharp emission line first rising above the spontaneous emission background and collapsing again under excessive current injection due to the self-heating of the laser. The line forms at 0.535 eV ($\lambda$ = 2.319 μm), inset Fig. 2a₁, first appears at a low injection current of 6 kA/cm², and rapidly develops into a sharp laser mode (inset Fig. 2a₂). This level of performance is equivalent to the one obtained with III-V DHS devices[40] at the time when the first CW electrical pumping lasing demonstrations were achieved in that material system[41]. A clearly measurable second emission line appears at a lower energy of 0.526 eV ($\lambda$ = 2.384 μm) for an injection current of 24 kA/cm², but it does not evolve into a strong emission. The laser is for this current already in a regime at which self-heating significantly limits emission. The difference in the emission energy of the two modes of ~9 meV is consistent with the expected WGM spacing for the 5 μm radius cavity, calculated to be ~9.4 meV. The laser emission collapses into spontaneous electroluminescence above 32 kA/cm² (inset Fig. 2a₃). A high-resolution measurement at 5 K under 12 kA/cm² current injection shows a sharp CW laser line with a full width at half maximum (FWHM) of 50 μeV, within the resolution bandwidth of the spectrometer of 25 μeV (Fig. 2b). Measurements under the same conditions but using a different, calibrated measurement setup with a lower resolution allows us to estimate a lower bound of the output power to 1 μW (Supplementary Fig. 1). The actual output power is expected to be higher due to the

fact that only scattered light is measured and the collection efficiency of the setup is thus limited.

The CW light-current (L-I) characteristic of the laser is presented in Fig. 2c for three temperatures. The abrupt change in the slope of the L-I curve indicates that stimulated emission becomes predominant over spontaneous emission. It is associated with the collapse of the spectral width of the emission, clearly indicating the onset of lasing (the full evolution of the FWHM with the injection current is shown in Fig. 2e and Supplementary Fig. 2). The injection current densities at threshold, $J_{th}$, are determined more precisely by using the second derivative method (Fig. 2d). They are 6.2 kA/cm², 8.5 kA/cm², and 9 kA/cm² at 5, 30, and 35 K, respectively. The current range over which lasing is observed shrinks as the temperature is increased. This is due to the interplay between ambient temperature and laser self-heating that impacts the gain. To better understand this aspect, the lattice temperature was estimated using the temperature dependence of the EL spectra from a large mesa LED (Supplementary Fig. 3) and an under-etched laser diode (Supplementary Fig. 4), both measured at low injection currents to avoid heating effects. In both cases, the Varshni fit of the bandgap temperature dependence results in almost identical fitting parameters, but the bandgap depends on the strain levels for different diode designs. The estimated lattice temperature of the active MQW layer for different current densities is presented in Fig. 2e, overlaid with the FWHM of the electroluminescence evolution. The lattice temperature seems to reach a high value of 104 K at a current density of 34 kA/cm², at which the laser emission collapses. Remarkably, the estimated lattice temperature is very close to the theoretical values of 110 K at which the carriers can escape from the QWs, due to the limited energy offset between SiGeSn barriers and GeSn wells (see Fig. 1d). Above this temperature, the carrier confinement fades out.

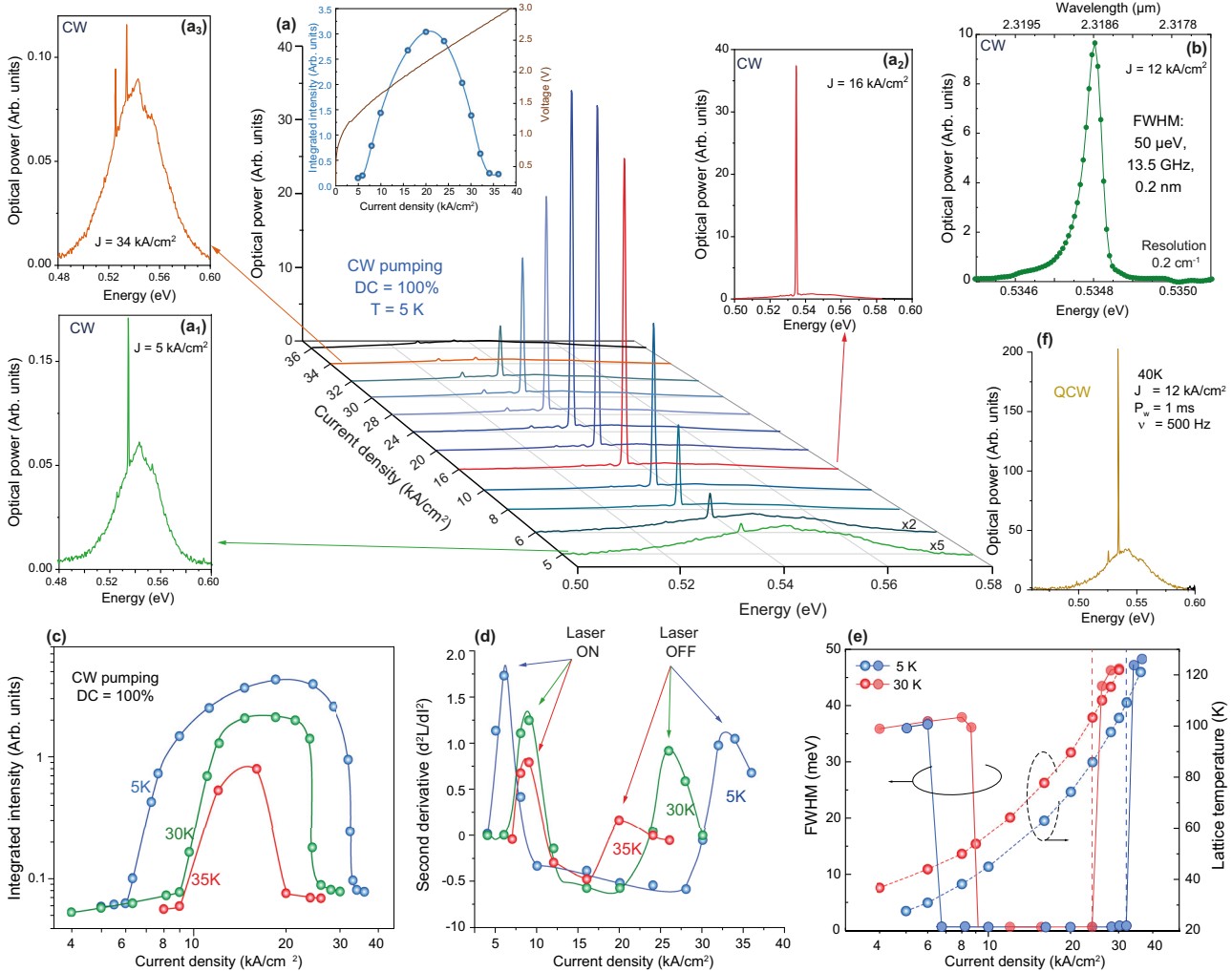

**Fig. 2 | Electrically pumped laser emission. a** EL spectra and L-I-V characteristic of a 5 μm radius diode showing a laser line emerging from the spontaneous emission background under CW operation at 5 K. Insets: spectra before threshold (a₁), after threshold (a₂), and after collapse of the laser line (a₃). **b** High-resolution spectrum showing an FWHM of 50 μeV for the main lasing mode. **c** L-I characteristics at 5, 30, and 35 K. **d** Second derivative of the L-I characteristics used to extract the laser ON and laser OFF thresholds. **e** Estimated lattice temperature vs. current pumping density overlaid with the FWHM evolution at 5 and 30 K. **f** Laser spectrum at 40 K under 1 ms, 500 Hz, DC = 50% current pulses.

CW lasing is the most challenging mode of operation to obtain, especially for low-gain materials, due to the self-heating of the device that raises the lattice temperature of the gain medium. Heating can then be reduced by quasi-continuous-wave (QCW) operation, in which the pump current is switched off for a limited amount of time, but maintained long enough in the on-state, e.g., milliseconds, to reach steady-state conditions. For the SiGeSn/GeSn MQW diode presented here, pure CW operation stops at 35 K. Switching over to millisecond QCW operation further increases the maximum lasing temperature. A laser spectrum taken at 40 K for a current pulse length of 1 ms at a frequency of 500 Hz, with a duty cycle (DC) of 50%, is shown in Fig. 2f.

**QCW and pulsed laser emission study**

The CW laser emission demonstrated above is a first step towards a monolithic integration of laser sources with SiPh for the emerging field of cryogenic photonics. The laser operation is in agreement with the electronic band structure of the designed heterostructure and with the optical properties of the laser cavity. However, for a deeper understanding of the laser behavior, two features, which give indications for future device performance improvement, can be further analyzed: (i) the temperature dependence of pulsed laser emission and (ii) the

relationship between laser properties and current pulse length. QCW behavior above 35 K can be used to get further insight into the characteristics of the laser.

As mentioned above, the heat dissipated during electrical pumping is one of the major factors leading to gain decrease. It has an impact on carrier energy and, thus, on the escape rate from the wells. Furthermore, heat reduces the material directness, resulting in higher scattering rates from the Γ- to the L-valley. It also increases the rate of non-radiative recombination processes, such as the Shockley-Read-Hall or Auger processes, that compromise the optical gain. Upon reducing the heat load by decreasing the electrical pulse length, the ambient temperature can be increased and the current density at threshold is reduced. In Supplementary Fig. 5, a set of 10 laser spectra is presented for current pulses ranging from 165 ns to 1 ms at a constant current density of 25 kA/cm², at 5 K. For a 5-μm-radius microdisk, the lasing threshold decreases from 6.2 kA/cm² for CW lasing operation down to 5.8 kA/cm² for 1 ms pulses and 5 kA/cm² for 100 ns pump pulses. The maximum ambient lasing operation temperature increases above 70 K for laser pulses below 1 μs and DCs below 10%. Temperature-dependent laser spectra for current pulses of 1 μs and a DC of 5%, taken at 1.5 × J_th, are presented in Fig. 3a.

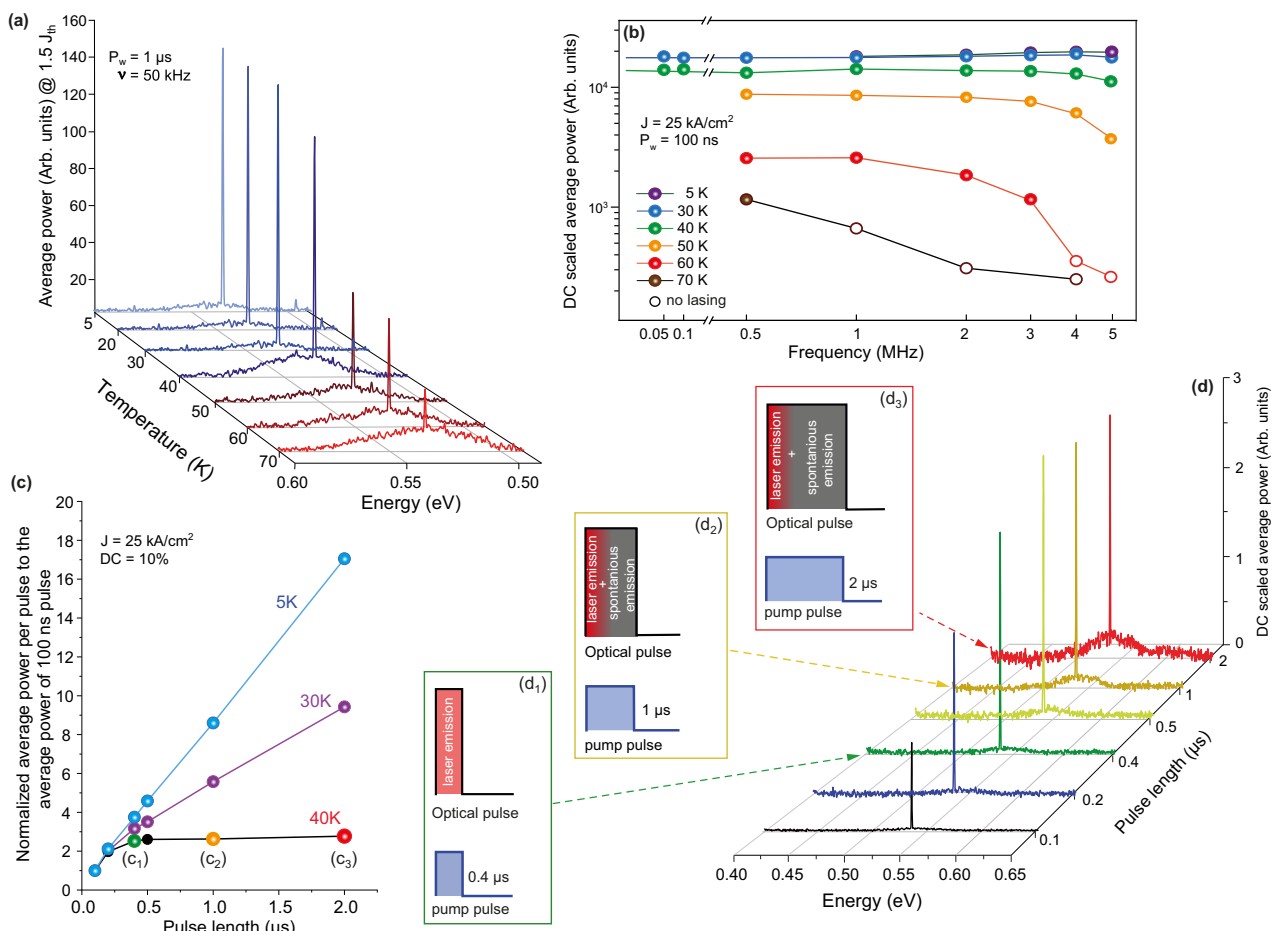

**Fig. 3 | Lasing under pulsed pumping. a** Temperature-dependent EL spectra taken at current densities of $1.5 \times J_{th}$ for a pump pulse of 1 µs. **b** Pulse frequency dependent average power for 100 ns pulses at different temperatures up to 70 K. The collected optical power is rescaled according to the duty cycle, to enable comparison between measurements. **c** Optical power per pulse normalized to that of the 100 ns pulse. The flattening of the curves indicates that the laser pulse length is shorter than the current pulse length. **d** Current pulse-dependent EL spectra taken at 40 K with a current density of 25 kA/cm² and a constant DC of 10% (corresponding to the black plot in (**c**)). The insets (**d₁–d₃**) schematically illustrate the relation between the length of the laser and current pump pulses.

A laser tunable over a wide range of current injection conditions gives, for the first time in group-IV photonics, the opportunity to study the stimulated emission behavior of GeSn semiconductors. The measured spectra, which are described so far, provide information on the spectral distribution of the signal intensity. However, they do not offer insights on the time scale of the laser pulses. The main question to answer now is whether the laser emission occurs over the entire current pulse duration at temperatures above 35 K. At low temperatures, at constant repetition rate and peak injection current, the averaged power increases linearly (slope $m = 1$) with the pump pulse length (see Supplementary Fig. 6). Consequently, at these temperatures, the laser emission duration equals the pump time duration, consistent with the laser's ability to operate in CW mode up to 35 K. Heat dissipation through the Ge pillar into the Si substrate is thus sufficient. Interestingly, up to 30 K, the heat dissipation through the Ge pillar is improved with increasing temperature, caused by the strong increase in the Ge lattice thermal conductivity. It reaches its maximum at 30 K, followed by a fast decrease of about 10 times at higher temperatures[42].

In structures with weak carrier confinement, as those used here, interesting laser dynamics appear at higher temperatures, where the energy of electrons increases, and heating effects play an important role. At 40 K, the laser emission is constant for 100 ns pulses up to the highest measured repetition rate of 5 MHz (see Fig. 3b). At 50 K, the laser emission drops slightly beyond 3 MHz. This indicates a

temperature rise due to non-sufficient cooling between subsequent current pulses. At 70 K, lasing stops above 1 MHz, which is a consequence of both: the reduction of the available optical gain and the increase of the thermal resistance of the Ge pillar, which slows down the cooling rate between pulses.

To evaluate the laser dynamics at 40 K, the optical emission is evaluated for different pump pulses at a constant peak current of 25 kA/cm² and a constant DC of 10%, so that the average electrical power is maintained (Supplementary Fig. 7). The averaged power per pulse relative to the emission for 100 ns current pulses is shown in Fig. 3c. At this temperature, in contrast to the curves at 5 and 30 K, the laser emission no longer increases linearly with the current pulse length, as would be expected when the lasing intensity stays constant over the entire duration of the current pulse. At 40 K, the average optical energy per pulse increases up to 400 ns current pulses and remains constant thereafter. This shows that laser action is maintained only during the first 400 ns, with spontaneous emission for the remaining current pulse duration, as schematically represented in the insets of Fig. 3d. This is also supported by the optical spectra shown in Fig. 3d, in which the emission background rises for longer current pulse durations. The optical emission is actually the superposition of a laser pulse of 400 ns and a spontaneous emission background during the rest of the increasing pulse duration. A similar analysis, presented in the SI, concludes that the laser pulse length reduces to 250 ns at

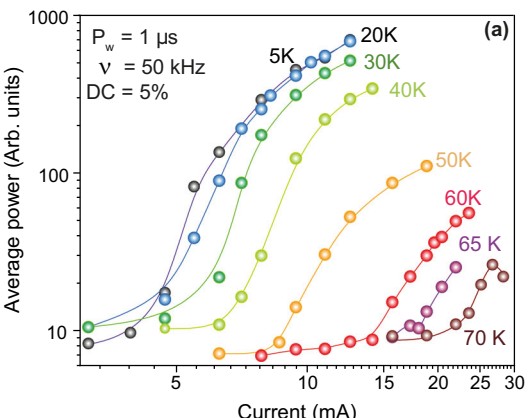

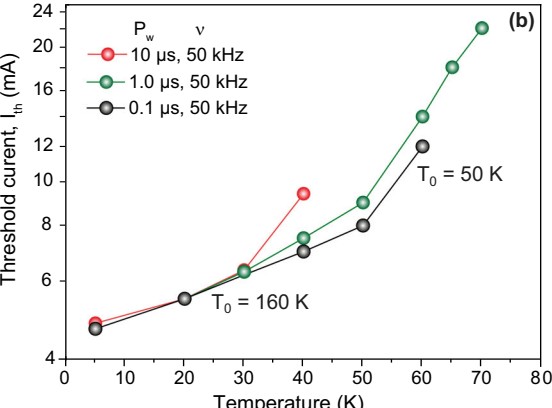

**Fig. 4 | Laser threshold. a** Temperature-dependent L-I characteristics for 1 μs pump pulses and a DC of 5%. **b** Laser threshold versus temperature. $T_0$ of 160 and 50 K are extracted at low and high temperatures, respectively.

60 K, while at 70 K, lasing occurs only during the first 100 ns of the pumping pulse.

## Temperature dependence of laser parameters

The laser threshold dependence on lattice temperature is an essential property of semiconductor lasers. Temperature-dependent L-I characteristics for 1 μs pulses at 50 kHz repetition rate are shown in Fig. 4a. Laser spectra, taken at 1.5 times the threshold current density, are already shown in Fig. 3a. The laser threshold has an $\exp(T/T_0)$ dependence on temperature, where the characteristic temperature of the laser, $T_0$, is a measure of its temperature sensitivity and $T$ is the laser operation temperature. $T_0$ is found to be 160 K for temperatures below 50 K and small pulse durations. Above that, $T_0$ decreases to 50 K, as seen in Fig. 4b. For longer pulses, the transition from high to low $T_0$ occurs at lower threshold temperatures due to self-heating of the laser. There are several factors that can contribute to this $T_0$ degradation. As the temperature rises, increased band filling and carrier thermalization eventually lead to the L-valley population. Consequently, the electron DOS suddenly increases. Thus, any further increase in Γ-valley electrons comes with a large overhead in L-valley electrons, leading to a runoff in the threshold current[31]. Reduced carrier confinement in the shallow wells and the temperature dependence of Auger recombination coefficients could also contribute to this. In particular, if it is assumed that the required carrier concentration continues to exponentially increase with a characteristic temperature of 160 K when Auger recombination becomes dominant, the required injection current would increase with the carrier concentration to the power of three, hence with an exponential dependence with three times smaller characteristic $T_0$, in line with what is observed here.

## Discussion

The laser emission sensitivity to temperature is attributed to weak quantization. The energy separation between quantized sub-bands in the CB or valence band (VB) is very small (1–2 meV), therefore, electrons and holes can easily transfer from one quantized state to another. Optical transitions, however, are stronger between electron and hole states with the same quantum number. Up to 35 K the laser can be operated in CW mode and the characteristic thermal energy $k_B T$ remains below the energy difference between the first and third sub-bands. This means that only the first two sub-bands contribute to gain. Quantization still significantly contributes to reducing the joint density of states (JDOS) below the photon energy, reducing the transparency carrier concentration and the required pump current. Above 50 K, four or more sub-bands are populated, implying that the gain medium behaves increasingly like bulk. However, even in the present MQW structure, the quantization is clearly essential. In the range where

lasing exists, the profiles of the spontaneous emission background for low and high current densities, shown in Fig. 2a₁ and a₃, are very similar (in shape, not intensity). This is another indication that the emission is based on 2D, not 3D DOS, because in the case of a 3D DOS, the spontaneous emission profile would shift with the carrier density. Strong quantization with a larger spacing between quantum states would enable to increase the temperature and this can be achieved by narrowing the width and increasing the Sn content in the wells. However, a trade-off had to be found between sufficient quantization and the decrease of CB directness resulting from the higher quantization energy offsets seen by Γ-valley electrons. The MQW design, meaning the Sn content and the well width, must thus be adapted to the desired applications and material system.

Even though the threshold is still slightly higher than the 0.6 kA/cm² of comparable GeSn Fabry-Perot lasers[30], the MQW strongly improves laser performance compared to the only electrically pumped microdisk SiGeSn/GeSn/SiGeSn double heterostructure (DHS) laser reported to date[31]. The threshold current density decreases by a factor of 8, although the cumulative thickness of the gain material is decreased by a factor of 3. This proves that quantization and possibly reduced non-radiative recombination rates resulting from the high material quality inherent to a reduced Sn content have an impact on laser performance. Lasing in MQWs is achieved in the CW regime and any pulse width/frequency combination, while the DHS structure was only able to lase with very short current pulses. Such improvements were achieved even though the DHS laser had a larger Sn composition of 14 at.% and a larger under-etch of 1300 nm, which both increase the energy difference between the Γ- and L-valley, making the gain medium more direct. For the same under-etch of 900 nm as reported here, the lasing threshold is reduced about 12 times compared to the DHS. This underscores the advantage of carrier confinement in QWs and the DOS change from 3D to 2D. It also highlights that increasing the directness of the material is not the only relevant criterion. Finally, the laser has proven to be resilient to extended operation times. To collect the data presented here, one of the laser diodes was kept in the ON-state for a cumulative time of ~60 h with over 10 million ON/OFF switching events. No diode has shown performance degradation at any time during these measurements.

The laser's operational temperature presented in this paper could be increased by further heterostructure and cavity design optimization. The Sn/Si compositions and the well/barrier widths could be further optimized to offer a larger band-offset. This can be obtained by increasing the Sn content inside the wells and the Si content inside the barriers, or by using a GeSn_x/GeSn_y MQW (x < y) with y-x > 5%[18]. Moreover, QW dimension optimization, like reducing the width of the wells, will lead to a larger operation temperature provided the

directness of the material can be maintained with the help of the improved material compositions. In addition, the use of a ring cavity design has been shown to lower the threshold even further. The ring cavity optimizes the overlap of the WGM with the pumped area and more importantly prevents pumping of the central part from a disk cavity that does not contribute to the amplification of the WGM but increases the overall heat load of the laser[31]. Heat dissipation is an important aspect for GeSn lasers, taking into account that GeSn becomes a good thermoelectric material with increasing Sn content[43]. Future works should also adopt strain engineering methods that are well established for Si technology, such as adding stressor layers, e.g., silicon nitride (SiN), to induce tensile strain in the active region[18]. From an integration perspective, the evanescent coupling of the lasing mode into a Ge or SiN waveguide or the use of a Fabry-Perot cavity with an increased Sn content of the GeSn buffer to provide relaxed or even tensile strained growth of the active layers are logical future steps.

In summary, a CMOS-compatible GeSn/SiGeSn multiple quantum-well heterostructure laser for cryogenic Si-photonics operation has been demonstrated. Its electrical characteristics show clear rectifying behavior. Laser operation is obtained from the short, 100 ns pulses to the CW range, with a linewidth of 15 GHz. Carrier confinement, quantization-induced change to a 2D-DOS in the gain material, and low contact resistances result in a much-reduced threshold current of only 4 mA. This is eight times lower compared to a previously reported thick GeSn laser device, which was able to operate only under very short pulse pumping conditions. This work opens up a path for the integration of GeSn-based lasers in large-scale PICs for emerging cryogenic systems in both classical and quantum-computing applications.

## Methods

### Fabrication

For the laser diodes, a design comprising an under-etched optical cavity (Fig. 1b) has been chosen. The fabrication was carried out using only CMOS-compatible processes and materials. It features a maximum temperature of 150 °C which is definitely below the growth temperature of the gain material, e.g., 300 °C. The GeSn mesa was structured down to the doped Ge-VS using inductive-coupled plasma reactive-ion-etching (ICP-RIE) with HBr radicals. After mesa structuring, an isotropic under-etch with F-based gas was used to form the microdisk cavity. During this step, the formation of a $SnF_x$-layer prevented Sn-containing layers from being etched. Diodes were passivated and electrically isolated by an 800 nm thick $SiO_2$ layer, which was deposited by plasma-enhanced CVD. Etching of the contact windows was done by RIE with $CHF_3$ gas, and a short wet-chemical etch in buffered HF. The metal contact module is realized by sputtering a 50 nm Ti diffusion barrier followed by 2 μm of Al, which were then structured using ICP-RIE with $Cl_2$ as etchant.

### Optical characterization

Laser spectra of MQW micro-ring diodes were acquired with a Bruker VERTEX 80 v FT-IR spectrometer. Pulsed emission (<50 kHz) was measured in the Step-Scan Mode, which detects the modulated signal intensity with a lock-in amplifier to achieve a higher signal-to-noise ratio, and hence better visibility for low-intensity signals. CW and high-frequency spectra were acquired in the continuous FT-IR scan mode with the integration of 80 spectra to reduce the background noise level. Finally, diodes were pumped using a Tektronix AFG function generator.

## Data availability

The experimental data generated in this study have been deposited in the FigShare database under the accession code https://doi.org/10.6084/m9.figshare.27242169.v1.

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

## Acknowledgements

The authors acknowledge financial support from the German Research Foundation (DFG) under Projects No. 299480227 "SiGeSn Laser for Silicon Photonics" (L.S., B.M., V.K., and T.L.) and 431314977/GRK2642 (L.S.), and from the European Commission for the LASTSTEP Project under grant agreement 101070208 (O.C. and T.L.).

## Author contributions
J.M.H. fabricated the Ge/Si substrates. V.K. and O.C. performed the epitaxial growth of GeSn/SiGeSn heterostructures. L.S., D.S., A.B., and M.O. fabricated the LEDs and the laser diodes. D.S., G.C., and O.C. carried out the TEM, XRD, RBS measurements, and ECV analysis. Z.I., B.M., and O.C. performed the band structure simulations. T.L. and L.S. performed all the laser measurements and analysis. D.B. and A.T. planned the experimental measurements. G.C., D.G., J.S., and D.B. supervised the experiments. A.T., G.C., J.W., and D.B. coordinated the data interpretation. All authors discussed the results and worked on the manuscript.

## Funding

## Competing interests
The authors declare no competing interests.
