## [Peer Review File · Nature Communications]

REVIEWER COMMENTS

Reviewer #1 (Remarks to the Author):

The authors reported CW lasing from SiGeSn/GeSn MQWs by electrical pumping at low temperatures. They also revised the manuscript substantially to address all issues raised by referees. The output power estimation of the lower bound of 1uW is mentioned, and the potential application in quantum and sensing domain at MIR were discussed. The RPCVD process is promising for the CMOS compatibility and the demonstration using a 200mm wafer proves the potential for future monolithic integration. I expect practical challenges to introduce these lasers together with front-end CMOS FETs, but the idea to use GeSn channel together with these lasers is interesting possibility, as suggested by the authors. It is indeed important to demonstrate the potential capability, and the achievement of this work is substantial.

Overall, the revised manuscript is well-written, and the analysis is quite comprehensive. I recommend publishing this paper.

Reviewer #2 (Remarks to the Author):

This paper reports the growth, fabrication, and characterization of a cryogenic CW GeSn micro-disk laser. GeSn laser on Si holds great promise to offer a monolithic integration solution which has been pursued for long time. It has been a viable competing technology with III-V on Si to achieve this goal. It seems that Sn based laser work has been made significant progress in the last a few years. The result reported in this manuscript potentially could be one step further to bring the material for the application. It seems that this manuscript previously was submitted to Nature Photonics. The authors have tried to address some review comments to improve the paper quality to ensure a better reading experience. Although not being able to see the complete review comments, I could make a guess based on the context of the conversation from the rebuttal letter. My general impression is that the result itself has reached a level to be published on Nature Communications but it will be nice if the authors could mark up their changes from their original Nature Photonics submission, which will make this review process much easier. Based on the results presented in this paper, I would recommend “acceptance” with major revision. Some comments might be similar to the previous reviewers.

1. Please mark your revision for this review and last Nature Photonics review with two different colors for next submission assuming that this paper most likely will be accepted.
2. I like the discussion between the authors and the reviewers. Some of those rationale thinking should be incorporated in the manuscript, for the benefits of audiences to understand the overall field development.

3. For the response to Reviewer #2, it inspires me about the benchmark with III-V. Group-IV laser is still in the early phase of its development, I am wondering if this result should be benchmarked with early III-V laser development work, for example, the first electrically injected III-V laser work? Some comparison will be helpful. Other than this, I am also wondering if the reviewer #2 might mean "GaSb based lasers" as the GeSn based lasers and GaSn based lasers work in the similar wavelength range. So some comparison will be helpful. (Note, people in general could understand the material challenges of group-IV materials to make lasers, the authors should not worry about the difference of the key values, such as threshold density. At least it is not the intention of this reviewer to use III-V data to disqualify the publication of this paper. But readers would be interested in knowing on that.)

4. "Benchmark with GeSn laser itself"

It does not seem that the authors compared their results with F-P laser results. This should be added. Readers should be able to understand the difference due to different cavity characteristics. But in semiconductor laser field, there are standard protocols to compare the material development progress. Using disk cavity is not an ideal approach to evaluate the material potential, the authors have acknowledged this in their response letter. So including some discussions will be helpful for people to understand where the technology stands.

5. Pathway to increase the operation temperature

This perhaps is a major concern for me.

The authors argued a lot regarding their active region design. I could not fully understand their logic. The carrier confinement is so low and the injection level is very high, which means the material gain is low and modal gain is also low for this design. So the potential for this architecture to achieve higher operating temperature is very limited, which could also mean that the material has very limited potential in future to compete with III-V for optical integration application.

The authors should elaborate more about their justification that lasing is due to gain from 2D DOS not 3D DOS.

In addition, I would suggest that the authors dedicate a discussion section regarding the potential pathway to further increase the operation temperature to room temperature. By reading the relevant publications, it seems that optically pumped GeSn bulk lasers could operate to RT, so in principle electrically injected should also be?

Reviewer #3 (Remarks to the Author):

The noteworthy result is that this is the first CW electrical pumped operation of a group IV laser.

Significance: this is significant as it has not been achieved with any group IV laser to date. It could have major impacts to silicon photonics by allowing monolithically integrated lasers in the future.

Does the work support the conclusions: yes - the data and experiments are quite detailed.

Any flaws - only very minor points which I state below.

Is the methodology sound? Yes

Is there enough detail provided in the methods for the work to be reproduced? Yes

A few corrections need to be completed before the manuscript can be published:-

1. page 3 - line 66 - "most promising" is too strong considering how immature most of these areas are. Remove the word "most".
2. page 3 - line 75 - InSb lasers are available from 3 to 5 μm e.g. A. Gassenq et al., "InAs/GaSb/InSb short-period super-lattice diode lasers emitting near 3.3 μm at room-temperature" *Elec. Lett.* 45, 165 (2009) and W. Y. Ji et al., "InAs/GaSb/InSb short-period super-lattice diode lasers emitting near 3.3 μm at room-temperature" *APL* 113, 232103 (2018). This statement needs to be corrected.
3. page 6 - line 142 - "Schematic" is an adjective in English which requires a noun so "Schematic diagram" or "Schematic figure". The authors should correct.

Response to the reviewers' comments

The authors thank the reviewers for the very high appreciation of the technical and scientific quality of the manuscript. Their comments and suggestions are incorporated and highlighted in the new manuscript version that we resubmit.

Reviewer #1 (Remarks to the Author):

The authors reported CW lasing from SiGeSn/GeSn MQWs by electrical pumping at low temperatures. They also revised the manuscript substantially to address all issues raised by referees. The output power estimation of the lower bound of 1uW is mentioned, and the potential application in quantum and sensing domain at MIR were discussed. The RPCVD process is promising for the CMOS compatibility and the demonstration using a 200mm wafer proves the potential for future monolithic integration. I expect practical challenges to introduce these lasers together with front-end CMOS FETs, but the idea to use GeSn channel together with these lasers is interesting possibility, as suggested by the authors. It is indeed important to demonstrate the potential capability, and the achievement of this work is substantial.

Overall, the revised manuscript is well-written, and the analysis is quite comprehensive. I recommend publishing this paper.

Answer: Thank you for this very positive evaluation of our work! It is seldom to get such a “laudatio” as review from a Nature journal.

Reviewer #2 (Remarks to the Author):

This paper reports the growth, fabrication, and characterization of a cryogenic CW GeSn micro-disk laser. GeSn laser on Si holds great promise to offer a monolithic integration solution which has been pursued for long time. It has been a viable competing technology with III-V on Si to achieve this goal. It seems that Sn based laser work has been made significant progress in the last a few years. The result reported in this manuscript potentially could be one step further to bring the material for the application. It seems that this manuscript previously was submitted to Nature Photonics. The authors have tried to address some review comments to improve the paper quality to ensure a better reading experience. Although not being able to see the complete review comments, I could make a guess based on the context of the conversation from the rebuttal letter. My general impression is that the result itself has reached a level to be published on Nature Communications but it will be nice if the authors could mark up their changes from their original Nature Photonics submission, which will make this review process much easier.

Based on the results presented in this paper, I would recommend “acceptance” a with major revision. Some comments might be similar to the previous reviewers.

1. Please mark your revision for this review and last Nature Photonics review with two different colors for next submission assuming that this paper most likely will be accepted.

Answer: The Nat Comm submission is an improved version of the Nat Phot manuscript, as requested by the editors. The Nat Comm manuscript is under review here and not any

previous versions. We will provide a marked revision for the manuscript in relation with the Nat Comm reviewers comments.

2. I like the discussion between the authors and the reviewers. Some of those rationale thinking should be incorporated in the manuscript, for the benefits of audiences to understand the overall field development.

Answer: We are pleased to hear that our correspondence with the former Nat Phot reviewers is appreciated. It is the scope of the peer-review process to stimulate fair scientific discussions that in most cases lead to technical and scientific improvements of the manuscript. The transferred manuscript to Nat Comm, as Reviewer 1 already mentioned above, “... revised the manuscript substantially to address all issues raised by [the NP] referees” and is in accordance with the request from both NP and NC editors.

3. For the response to Reviewer #2, it inspires me about the benchmark with III-V. Group-IV laser is still in the early phase of its development, I am wondering if this result should be benchmarked with early III-V laser development work, for example, the first electrically injected III-V laser work? Some comparison will be helpful. Other than this, I am also wondering if the reviewer #2 might mean “GaSb based lasers” as the GeSn based lasers and GaSn based lasers work in the similar wavelength range. So some comparison will be helpful. (Note, people in general could understand the material challenges of group-IV materials to make lasers, the authors should not worry about the difference of the key values, such as threshold density. At least it is not the intention of this reviewer to use III-V data to disqualify the publication of this paper. But readers would be interested in knowing on that.)

Answer: Thank you for giving the chance to add some kind of III-V comparison. We had such comparisons before and we took them out because it generally leads to complains from III-V reviewers regarding poorer values of key parameters, as the reviewer just mentioned above. As consequence we have tried now to avoid “comparisons” with 70 years of III-V lasers development that is not fair. We wanted to show a new development in the field of group IV lasers and the importance of the data by its own. A CW laser on Si is not meant to compete/attack or take the place of the high-performance, market available III-V lasers. We have added now some lines regarding the comparison between our CW laser and the first CW electrically pumped III/V laser. We also mention and added a short comment to the GaSb-based lasers.

4. “Benchmark with GeSn laser itself”

It does not seem that the authors compared their results with F-P laser results. This should be added. Readers should be able to understand the difference due to different cavity characteristics. But in semiconductor laser field, there are standard protocols to compare the material development progress. Using disk cavity is not an ideal approach to evaluate the material potential, the authors have acknowledged this in their response letter. So including some discussions will be helpful for people to understand where the technology stands.

Answer: The pulsed (one pulse width only!) GeSn laser published previously is presented in the manuscript. We have added a new line indicating a lower threshold was obtained in pulse mode for FP GeSn lasers but no CW emission. This is here reported for the first time. We definitely agree that the FP design is a better choice for single laser diode characterization, like direct measure of the output power, or for easier light coupling into

waveguide. However, it is not a priori better for proof of principle of laser emission and for gain material study, as this manuscript also shows.

It is not easy and maybe not even fair to compare the only two available GeSn laser results based on cavity design only. Why does a FP design result in a lower threshold, meaning better laser efficiency, but not in CW operation which is a characteristic of low losses-high gain? Is the FP homogeneously pumped and lases over the whole FP length/width? If not, the calculation of the current density is wrong and the threshold is in reality much larger. Or is in our MQW material the gain much larger than the bulk GeSn in the published work? We do not know the material, the fabrication and the characterization of the gain material of our colleagues and it is no good science to speculate about their work. We can only compare the reported naked values. When more laser will be available in literature than a fair comparison can be made. Now, it is too early and not appropriate in our opinion.

5. Pathway to increase the operation temperature

This perhaps is a major concern for me.

The authors argued a lot regarding their active region design. I could not fully understand their logic. The carrier confinement is so low and the injection level is very high, which means the material gain is low and modal gain is also low for this design. So the potential for this architecture to achieve higher operating temperature is very limited, which could also mean that the material has very limited potential in future to compete with III-V for optical integration application.

Answer: First of all, again, we do not see the development of GeSn devices in competition with III-V lasers. This view of a new material development as a war with other material systems is, destructive, non -scientific and we reject it with the strongest words. GeSn may, if mature enough at some point, have applications in a wavelength range that is not well covered by the classical III-V materials. The easiest and already tested applications are in sensing applications like gas sensing, liquid etc. The benefit is not that it is much better than the III-V but offers compatibility to Si, meaning easier fabrication together with sufficient sensitivity (not better).

The authors should elaborate more about their justification that lasing is due to gain from 2D DOS not 3D DOS.

Answer: It is well known that the MQW lasers are better than bulk in terms of the ratio between threshold current and active volume. Regarding the justification (prove?) that we have an MQW, we have added a new paragraph in the manuscript.

In addition, I would suggest that the authors dedicate a discussion section regarding the potential pathway to further increase the operation temperature to room temperature. By reading the relevant publications, it seems that optically pumped GeSn bulk lasers could operate to RT, so in principle electrically injected should also be?

Answer: We have done this to the best of our knowledge based on published information. Yes, the bulk lasers reach RT. The electrical lasers must be able to reach RT, too, if the joule heating will be below the optical heating. The formation of low sheet and contact resistance metal contacts as well as optimum p and n doping are studied. Note that the lasers presented

in this work have no p-type doping in the classical way. The doping here is material intrinsic via crystalline point defects. This is mentioned in the manuscript. For MQW lasers there is no RT operation published. We had obtained recently optically pumped RT MQW lasers but these developments cannot be reported and commented here. However, the Discussion part that includes also an outlook was re-read and improved.

Reviewer #3 (Remarks to the Author):

The noteworthy result is that this is the first CW electrical pumped operation of a group IV laser.

Significance: this is significant as it has not been achieved with any group IV laser to date. It could have major impacts to silicon photonics by allowing monolithically integrated lasers in the future.

Does the work support the conclusions: yes - the data and experiments are quite detailed.

Any flaws - only very minor points which I state below.

Is the methodology sound? Yes

Is there enough detail provided in the methods for the work to be reproduced? Yes

A few corrections need to be completed before the manuscript can be published:-

1. page 3 - line 66 - "most promising" is too strong considering how immature most of these areas are. Remove the word "most".
2. page 3 - line 75 - InSb lasers are available from 3 to 5 μm e.g. A. Gassenq et al., "InAs/GaSb/InSb short-period super-lattice diode lasers emitting near 3.3 μm at room-temperature" Elec. Lett. 45, 165 (2009) and W. Y. Ji et al., "InAs/GaSb/InSb short-period super-lattice diode lasers emitting near 3.3 μm at room-temperature" APL 113, 232103 (2018). This statement needs to be corrected.
3. page 6 - line 142 - "Schematic" is an adjective in English which requires a noun so "Schematic diagram" or "Schematic figure". The authors should correct.

Answer: Thank you for your careful read of our manuscript and for the appreciation of our work. We have corrected the mechanical deficiencies, corrected the information and added few new references. We believe that now our manuscript reads better.

REVIEWERS' COMMENTS

Reviewer #1 (Remarks to the Author):

I have carefully read the revised manuscript together with the replies to reviewers. I think the authors addressed all issues raised and I believe the lasing from group-IV materials were successfully achieved. I recommend the publication of this paper.

Reviewer #2 (Remarks to the Author):

The authors made efforts to revise the manuscript to address all questions and concerns raised by different reviewers.

I recommend acceptance of the manuscript.

Reviewer #3 (Remarks to the Author):

I am happy with the changes the authors have made and the manuscript is in my opinion ready to be published.